# The Influence of the Pharmacists’ Training on the Quality and Comprehensiveness of Professional Advice Given in the Field of Inhalation Techniques in Community Pharmacies in Poland

**DOI:** 10.3390/ijerph19042071

**Published:** 2022-02-12

**Authors:** Magdalena Waszyk-Nowaczyk, Weronika Guzenda, Patrycja Targosz, Michał Byliniak, Beata Plewka, Piotr Dąbrowiecki, Michał Michalak, Aneta Bąbol, Karol Szapel, Marzena Bielas, Jerzy Żabiński

**Affiliations:** 1Pharmacy Practice Division, Chair and Department of Pharmaceutical Technology, Poznan University of Medical Sciences, 6 Grunwaldzka Street, 60-780 Poznan, Poland; mwaszyk@ump.edu.pl (M.W.-N.); beataszukalska@gmail.com (B.P.); 2Student’s Pharmaceutical Care Group, Department of Pharmaceutical Technology, Pharmacy Practice Division, Poznan University of Medical Sciences, 6 Grunwaldzka Street, 60-780 Poznan, Poland; patrycjatargosz@gmail.com; 3Polish Pharmaceutical Chamber, 77 lok. 6 Żeromskiego Street, 01-882 Warsaw, Poland; michal@byliniak.com; 4Department of Infectious Diseases and Allergology, Military Institute of Medicine, 128 Szaserów Street, 04-141 Warsaw, Poland; piotr.dabrowiecki@wp.pl; 5Chair and Department of Computer Science and Statistics, Poznan University of Medical Sciences, 7 Rokietnicka Street, 60-806 Poznan, Poland; michal@ump.edu.pl; 6Lindsay & Gilmour Pharmacy, 5 Oliver Park, Hawick, Roxburghshire TD9 9BG, UK; anet.babol@gmail.com; 7Department of Physiotherapy, Poznan University of Medical Sciences, 11 Smoluchowskiego Street, 60-179 Poznan, Poland; kontakt@karolszapel.com; 8Department of Family Medicine, Poznan University of Medical Sciences, 49 Przybyszewskiego Street, 60-355 Poznan, Poland; mbielas@ump.edu.pl; 9Department of Organic Chemistry, Medical University of Warsaw, 1 Banacha Street, 02-097 Warsaw, Poland; jzabinski@o2.pl

**Keywords:** pharmaceutical care, inhaler education, community pharmacy

## Abstract

Background: Following the example of other countries, it is very important to educate patients on the correct use of inhalers by properly trained healthcare professionals, including pharmacists. Objectives: The aim of the study was to assess the quality and comprehensiveness of professional advice given by pharmacists on the use of inhalers, which was determined by the pharmacists’ level of training. Methods: The study was conducted from June 2019 to March 2020. 150 pharmacists from Poznan and Warsaw (Poland) were involved. Before the study began, the professional education of 240 pharmacists was conducted in Warsaw to implement standard operating procedures. The study used the model of a mystery shopper. Results: The conversation with a trained pharmacist lasted on average 5.5 min, with an untrained one—3.0 min (*p* < 0.0001). Placebo inhalers were used more often by trained pharmacists during patients’ education (*p* < 0.0001). Moreover, 10.3% of untrained pharmacists did not provide any education. Additionally, untrained employees’ quality of advice was assessed on an average of 3.5 points, while trained ones—7.6 points (*p* < 0.0001). Conclusions: This study has shown that there is a need for professional training among pharmacists in Poland, which translates into better patient education in the field of inhalation techniques.

## 1. Introduction

Recently, the conventional view of the pharmacist as mainly a person who is dispensing drugs in a pharmacy, has changed significantly in highly developed countries. Undoubtedly, this group of professionals plays an important role in the healthcare system, and the changes in the perception and understanding of it are more and more noticeable. Pharmacists are involved in the process of health promotion and education during their work [1,2]. Pharmaceutical care, as a responsible provision of drug therapy for the purpose of achieving definite outcomes that improve a patient’s quality of life [3], is successfully implemented in many countries around the world, such as Great Britain, the Netherlands, Norway, Finland, the United States. Many patients experience multiple benefits from this kind of service. It can improve the patient’s health, as well as the functioning of the entire health care system [4]. Patients suffering from asthma and chronic obstructive pulmonary disease (COPD) have the opportunity to take advantage of inhaler technique education in community pharmacies. It is standard practice for appropriately trained pharmacists to provide consultations regarding prescribed medications and their side effects, self-management of the disease, and instructions on how to use an inhaler [5].

Asthma and COPD are complex, multifactorial diseases that are often fluctuating in severity over time and require that patients receive support and a comprehensive education, the quality of which can affect the patient’s entire life. Good results in treatment depend on the effectiveness of medications and, most importantly, on the correct use of inhalers by patients. Education and training can be effectively delivered by a variety of health professionals, such as physicians, nurses, or pharmacists. Many of the therapeutic problems are caused by an incorrect inhaler technique. The most popular errors are, e.g., not breathing out before inhalation or not holding the breath for about 10 s after inhalation. It can lead to poor asthma control as well as numerous complications like oral thrush when the patient is not rinsing the mouth after corticosteroid inhalation [6]. Pharmacists are in an excellent position to educate such patients because they are in a highly accessible medical profession [7,8]. Very often only after training in community pharmacy, patients are able to demonstrate the correct inhaler technique [9]. It is worth mentioning that a pharmacist’s empathy and accurate questioning are very important for effective interviewing and providing professional advice for patients in a community pharmacy.

Following the example of other countries and the treatment effects they achieved, it is very important to educate patients on how to use inhalers properly. In Poland, there is a need to create appropriate procedures for counseling. There are no standards and guidelines for pharmacists to cooperate with patients in pharmaceutical care. Moreover, after graduation, pharmacists may participate in specialization courses, postgraduate studies, or thematic courses which expand general knowledge. Unfortunately, none of them introduce standard procedures in practice. Rarely do courses offer the form of workshops [10].

Therefore, the aim of the study was to implement the standard operating procedures for patients’ inhaler technique education and to assess the quality of professional advice given by pharmacists, which was determined by the pharmacists’ level of training, in the setting of community pharmacies in Poznan and Warsaw (Poland). Furthermore, the study verified whether the interview conducted by the pharmacist was sufficiently insightful. The study also assessed how comprehensive and detailed the patient’s education was in the technique of inhalation.

## 2. Materials and Methods

The main study was conducted from June 2019 to March 2020. Additionally, in October, December 2019, and March 2020, the professional education of 240 willing to participate pharmacists (86.3% women, 13.7% men) was conducted in Warsaw, by an experienced interdisciplinary team consisting of pharmacists and physicians (authors of the study), in order to increase the knowledge and implement standard operating procedures for patients’ inhaler technique education. It was the first practical training on the use of inhalers organized in Poland, in cooperation with the University of Medical Sciences and the Pharmacy Chamber in Warsaw. The training lasted 8 h and was divided into theoretical (4 h lectures about the latest asthma and COPD recommendations [11]) and practical parts (4 h workshops about the use of inhalers and standard operating procedures (SOP) published by the University of Medical Sciences and Pharmacy Chamber in Warsaw [12]). Each course consisted of about 40 participants, and during the workshop, the group of pharmacists was divided in half. Six training sessions were conducted. The study assessed how the training of pharmacists affects the quality of services and the level of professional pharmaceutical advice they are providing. Pharmacists were trained to an SOP and then assessed on how well they educated patients afterwards. The inclusion criteria for selecting the study group were employed in a community pharmacy as a pharmacist. One hundred and fifty polish pharmacists in Poznan (*n* = 80, 88.7% women, 11.3% men) and Warsaw (*n* = 70, 90.0% women, 10.0% men) were involved in the mystery shopper study. These pharmacists were divided into 2 groups: trained (43 people; 88.4% women and 11.6% men) and untrained (107 people; 89.7% women and 10.3% men). Forty-three pharmacists from 240 trained were selected randomly from the list of people who had undergone the course (Figure 1).

The study used the model of a “mystery shopper” (MS) [13]. This technique assumed that the trained researcher has taken the role of a patient, suffering from asthma in this case. The role was portrayed by the second researcher with previous experience in the MS method. The MS conducted a depth observation each time, assessed the scope of the information and the quality of the advice provided, and evaluated the competence of the pharmacist. During the visit to every community pharmacy, the researcher asked the pharmacist to demonstrate how the prescribed inhaler should be used. Each time, salbutamol pMDI (Pressurized Metered Dose Inhaler) was used. The conversation with the pharmacist was standardized according to the structured scenario. After leaving the pharmacy, particulars concerning the pharmacist’s advice and the scope of information provided to the MS about the inhaler technique were entered into a form based on the World Health Organization protocol [14]. This checklist included also issues related to basic questions asked by the pharmacist during the MS visit, e.g., for whom the drug was intended, had the drug been used previously, or the reason for prescribing the drug by the physician. The document was divided into two parts. The first one pertained to the questions that should be addressed to the patient who is going to be using the inhaler while they are in the pharmacy. The second part of the form was about the correct inhalation technique using the pMDI. The pharmacist educating the patient was awarded one point for each activity that was performed, from a list containing the description of the correct inhalation technique. It was possible to obtain a maximum of 9 points for inhalation activities. The document also contained information about the type of education that was provided, e.g., whether the pharmacist performed the training with the use of a placebo inhaler or solely with the use of verbal instruction. Moreover, it contained a subjective assessment of the visit, by the individual MS evaluation, using a point scale from 1 to 10. 

All pharmacists were informed during the training about the visits of mystery patients to their community pharmacy and the possibility of inclusion in the research before the study. All participants were clearly familiar with their role in this study (purpose, procedures, risks, benefits, etc.) and had the chance to ask clarifying questions if necessary. The participant understood what the research was and what they were consenting to, and the pharmacists provided informed consent to participate in the study. The collected data were securely stored in the Department of Pharmaceutical Technology, Pharmacy Practice Division at Poznan University of Medical Sciences. The study was conducted in accordance with the Declaration of Helsinki, and the protocol was approved by the Ethics Committee of Poznan University of Medical Sciences (602/19).

The Statistica PL 12 (StatSoft Polska Sp. z o.o., ul. Kraszewskiego 36, 30-110 Kraków, Poland) package was used to perform the statistical analysis. The correlations between analyzed nominal data were performed by chi-square test of independence. All statistical analyses were performed at *p* < 0.05.

## 3. Results

A properly conducted educational intervention should contain basic questions about the characteristics of the patient’s disease and the circumstances of drug administration. The survey indicated that 55.8% of trained pharmacists considered it necessary to ask the MS the question “for whom the drug was intended”, as opposed to untrained people, where this topic was not mentioned in 90.6% of cases (*p* < 0.0001). Moreover, 79.1% of trained pharmacists checked if the drug had been used previously, but 74.8% of the untrained people did not ask about this issue (*p* < 0.0001). More than half of the trained people (55.8%) asked about the reason for prescribing the drug by the physician, while amongst the untrained pharmacists, it was only 8.4% (*p* < 0.0001). The conversation with trained pharmacists lasted on average 5.5 min, and with an untrained person, just over half as long, 3.0 min (*p* < 0.0001). Comparing the results of the patient education that was carried out in the community pharmacy by trained and untrained pharmacists in Poland, the differences are clearly noticeable, as shown in Table 1. Out of the 9 elements of the correct inhalation technique, the trained pharmacists most frequently paid attention to inserting the mouthpiece in the mouth and ensuring a tight seal with the mouth (*p* < 0.0001) and pressing the inhaler immediately after starting to inhale through the mouth in order to release a dose of the drug (*p* = 0.0259). They also very often indicated the need to remove the mouthpiece cover (*p* < 0.0001). Furthermore, 81.4% of the trained subjects reported holding the inhaler upright (*p* = 0.0049) and performing a deep exhalation outside the inhaler before inhaling (*p* = 0.0002). Untrained pharmacists conducted significantly less extensive education about inhalation techniques. The study noticed that trained pharmacy employees much more often used training inhalers (51.2% for placebo inhalation) during patient education. Untrained people usually educate the patient solely through verbal instruction—63.5%; 10.3% of them did not provide any education for the patient (Table 2). After the visit, the quality of the advice was subjectively assessed by MS, on a scale from 1 to 10. In summary, untrained employees were assessed on an average of 3.5 points, while trained employees—7.6 points (*p* < 0.0001). The summary of the obtained points containing the steps of the correct pMDI inhalation during MS education, indicated that trained pharmacists obtained an average of 6.3 points out of the 9 that were possible, while the untrained ones acquired 4.8 pts (*p* < 0.0001).

## 4. Discussion

The training of pharmacists supervised by a multidisciplinary team is very important, especially with regard to the correct inhalation technique. It has been confirmed that such activity improves the everyday practice in community pharmacies [14]. Unfortunately, there are no suitable training programs in Poland, so the courses that were conducted in this study were pioneering. Their main effect was a better quality of patient care. Such activity is successfully implemented in many countries and delivers noticeable results in patient care [15]. Providing information on medications, as well as counseling patients on how to use the inhaler, are the most common elements of the pharmacist’s intervention [16]. 

The study showed that the training of pharmacists contributed to them asking patients more questions to help assess their health status. It seems obvious to ask, e.g., for whom the drug is intended. However, sometimes this basic question is omitted by some pharmacists [1]. The Polish study by Tomerska-Kowalczyk et al. showed that following appropriate standards for interviewing patients significantly influenced the number of questions and improved the quality of pharmaceutical consultations [17]. When counseling patients, a pharmacist should make sure that the drug has been used by the patient in the past. This is of particular importance for the further development of the advice. The counseling of a person who does not know how to use a drug will be different than that of a person for whom the drug has already been prescribed previously. Typically, a patient who has just received a new drug needs more time and attention in the pharmacy. However, even patients who demonstrate a sufficient inhalation technique may forget to take their medications regularly. The patients should understand why and how to use the drug correctly [18]. Trained pharmacists are three times more likely to ask the patient if the drug was used before in comparison to untrained ones. A similar study was carried out in Australia, where it was proven that trained pharmacists conducted ample education of patients, and its effects were clear and measurable [19]. Asking the patient about the reason for prescribing the drug allows for a more in-depth interview. In this way, a pharmacist can obtain information about the disease and provide professional advice, individually tailored for each patient. Conducting a sufficient and comprehensive interview about the patient’s health and education on the proper use of the inhaler, influences the consultation time. The conversation with the trained pharmacists lasted almost twice as long than with the untrained ones. It is a pharmacist’s duty to provide information and advice about the effects of drugs, and to make sure that the patient knows and understands the correct inhalation technique [20,21]. In the study by Basheti et al., it was shown that the majority of patients living in rural areas were unaware of their incorrect inhaler use, but the pharmacist’s educational intervention resulted in an improved inhalation technique [22]. It is worth highlighting that patients will search for health information from different sources such as the internet or magazines. For this reason, it is extremely important not to leave the patient to fend on their own, but to answer all of their questions and concerns [23].

According to Emeryk et al., effective education should last at least 6–10 min for one type of inhaler. All forms and techniques of education in this field are allowed, but a “live” demonstration is the most effective form of inhalation technique education [24]. This study showed that the education of a patient with the use of a demo version of the inhaler is not popular in Poland. More than half of the trained pharmacists used placebo inhalers, which significantly extended the duration of the consultation. This clearly influenced the improvement in the quality of the provided consultation. Unfortunately, it was confirmed that in many community pharmacies there was a lack of placebo inhalers, and patients were instructed verbally. It is worth mentioning that demonstration inhalers provide the best way for educating patients and are considered a standard in community pharmacies in many countries around the world [25,26].

The role of pharmacists in enhancing the skills of a patient in inhalation techniques is significant [22]. According to the research of Lindh et al., most of the mistakes made by patients are associated with the type of device used and an incorrect inhalation technique. The main problems were the lack of slow, deep exhalation before the start of inhalation, and an insufficient breath-hold time after drug application [27,28,29]. Based on the conducted study, it was shown that the trained pharmacists were able to conduct a more detailed consultation than untrained ones. Polish pharmacists are burdened with many administrative functions in a community pharmacy. Despite their training, they still did not implement all of the aspects of correct inhalation. It was most likely associated with a lack of sufficient time to conduct a full training for the patient. The most frequently omitted details were: shaking the inhaler well, taking an interval between doses, and replacing the dust cap. It is worth mentioning that each element omitted in the inhaler technique may influence the effectiveness of the treatment; therefore, additional attention should be paid to provide patients with more detailed and reliable information. Furthermore, the research confirms that carrying out only one training is often insufficient for the patient to repeat all the activities correctly. It has been proven that it is good to reinforce the given advice three times to achieve full inhalation skills [30]. The quality of the provided education by trained pharmacists was much higher in the subjective MS assessment. Moreover, the scores obtained for the individual activities performed during inhalation education were significantly higher for these pharmacists. Publications indicate that qualified pharmacists have been providing this kind of education in many countries for years. A professional pharmacist consultation may bring many positive effects to the patient’s pharmacotherapy; therefore, their role is extremely significant. [31,32,33]. Patients using inhalers require detailed instructions. Pharmacists should consult patients not only when receiving the first inhaler but also during each subsequent visit. This provides an ideal opportunity to evaluate the correct use of the device especially when the patient has prescribed two inhalers that require varying administration techniques. This can cause additional difficulties for the patient. However, a pharmacist should first undertake appropriate training themselves, in order to be able to educate patients effectively and properly. Research shows that trained pharmacists obtain practical skills and broaden their knowledge, which has a significant impact on improving asthma and COPD control in patients. Without specific training, they will not have the required skills to provide an effective consultation to patients [9,11].

Diseases of the respiratory system are serious health problems, and asthma or COPD are classified in the categories of civilization diseases [11,34]. The health system in Poland is increasingly struggling with the underdiagnosis of these kinds of patients, who instead of specialists, seek information on the internet, especially during the COVID-19 pandemic, when access to physicians is limited. Poor access to professional care is associated with a high risk of underdiagnosis and perilous therapeutic decisions [35]. Pharmacists definitely play an important role in patient health care. They are pivotal in sign posting physicians/respiratory specialists/physiotherapists, etc., to tackle the issue of underdiagnosis. While other specialists closed their doors to patients during the pandemic, community pharmacies remained open despite the risks. Pharmacists are now the most accessible health professionals [36]. Unfortunately, unlike in many countries where pharmaceutical care has been functioning successfully for many years [37], in Poland, we do not have official guidelines for professional pharmacological consultations. Currently, pharmacists can prescribe inhalers to patients when there is a continuation of chronic treatment [9]. It should be emphasized that pharmacists continue to develop their skills constantly. Due to the methodology used, the study is repeatable and similar tests can be performed for further problem comparison. Cyclical training has a positive impact because pharmacists are able to refresh their knowledge and find motivation. A systematic review of effective inhalation techniques showed that the effect diminishes over time; therefore, it is important to conduct periodic training, not only for the benefit of patients but also for pharmacists [38]. Pharmacy students also have great potential. During university classes, they should attend as many practical exercises as possible; for example, with the use of simulated patients, which have been shown to have a significant influence on the development of adequate skills [39]. It is worth investing in the education of students and pharmacists since this has a significant impact on the effectiveness of a patient’s pharmacotherapy, and significantly reduces unnecessary healthcare expenses. The investment in education and training of pharmacists has been shown to improve the health of patients with chronic diseases and constitutes a cost-effective alternative [40].

## 5. Conclusions

This study has shown the need for professional training among pharmacists, which translates into better patient education. Research results show that workshop methods in educating pharmacists can bring good results in everyday practice with a patient in a pharmacy community. Unfortunately, there was no pharmaceutical simulation center at the time of the study in Poland. Since it is available this year, using standardized patients with smaller pharmacists’ groups will indeed improve the practice of inhaler education with much higher efficiency.

## 6. Limitation of the Study

This study included a relatively small number of subjects. This was mainly caused by the COVID-19 pandemic. The planned group was larger, but the pandemic caused significant limitations to the study. Moreover, the trained group might be, in general, more engaged and keener to educate patients well vs. the untrained group, and it is hard to know how much of the results were down to this difference between the groups. However, these results showed some trends and directions in the field of pharmaceutical care in Poland. More detailed studies and better evaluation methods of the quality of the performance for the participating, trained pharmacists are needed in the future. It is worth mentioning that this study was innovative and brought many necessary guidelines for the implementation of pharmaceutical care in Poland.

## Figures and Tables

**Figure 1 ijerph-19-02071-f001:**
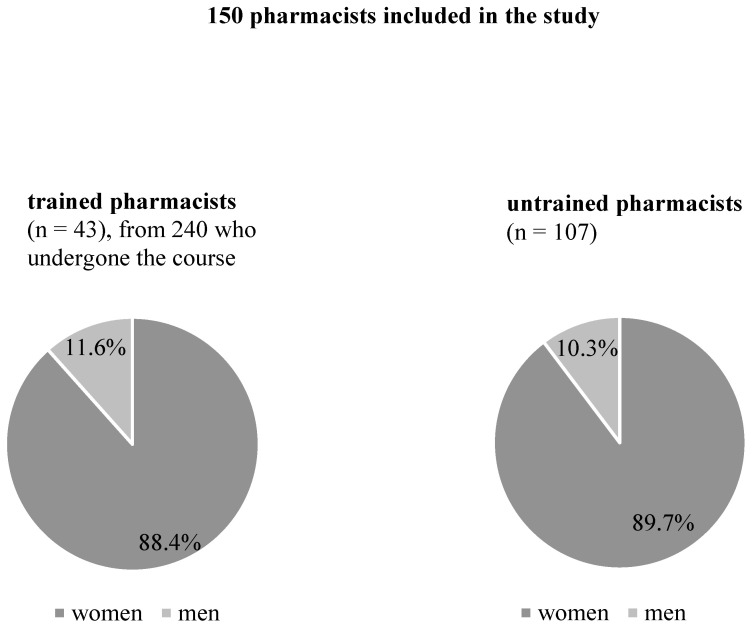
Pharmacists included in the study.

**Table 1 ijerph-19-02071-t001:** Information about the inhalation technique during patient education, determined by the pharmacist’s level of training (*n* = 150).

Inhalation Technique Education (pMDI)	Trained Pharmacist (*n* = 43)	Untrained Pharmacist (*n* = 107)
	*n* (%)	
Remove dust cap (*p* < 0.0001 *)	36 (83.7)	46 (43.0)
Shake inhaler well (*p* < 0.0001 *)	29 (67.4)	20 (18.7)
Keep upright (*p* = 0.0049 *)	35 (81.4)	61 (57.0)
Deep exhale (*p* = 0.0002 *)	35 (81.4)	51 (47.7)
Tight seal of the mouthpiece (*p* < 0.0001 *)	40 (93.0)	56 (52.3)
Press while inhaling (*p* = 0.0259 *)	40 (93.0)	83 (77.6)
Hold breath (*p* < 0.0001 *)	33 (76.7)	26 (24.3)
Interval between doses (*p* < 0.0001 *)	24 (55.8)	11 (10.3)
Replace dust cap (*p* < 0.0001 *)	20 (46.5)	5 (4.7)

* Statistically significant results at *p* < 0.05.

**Table 2 ijerph-19-02071-t002:** Type of conducted education, determined by the pharmacist’s level of training (*n* = 150, *p* < 0.0001 *).

Type of Education	Trained Pharmacist *n* (%)	Untrained Pharmacist *n* (%)
Placebo inhalation	22 (51.2)	2 (1.9)
Education using mystery shopper’s inhaler	14 (32.5)	26 (24.3)
Verbal education	6 (14.0)	68 (63.5)
Lack of education	1 (2.3)	11 (10.3)
TOTAL	43 (100.0)	107 (100.0)

* Statistically significant results at *p* < 0.05.

## Data Availability

The data presented in this study are available on request from the corresponding author.

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
