# Peer review of "The Influence of the Pharmacists’ Training on the Quality and Comprehensiveness of Professional Advice Given in the Field of Inhalation Techniques in Community Pharmacies in Poland"

_ijerph, 2022, doi:10.3390/ijerph19042071_

Round 1

Reviewer 1 Report

This is an interesting article. One rarely reads about Mystery Shoppers sent in by Pharmacy researchers.

I think the following information would add to this piece of work:

  1. Definition/reference of 'pharmaceutical care'. Line 55
  2. Poor inhaler technique leads to problems ; Line 65. Include reference and expand on problems
  3. Materials and methods: Some sections unclear. Lines 88-94.   Difficult to understand numbers??
  4. Materials and methods:  Suggest inclusion of chart/diagram to clarift different numbers provided e.g.Lines 102-108 unclear
  5. English: Line 121  i.e.    'for' whom was the drug intended
  6. Line 131.   'subjective' assessment - please explain
  7. Results: Line 147.  'professional counsel'?  Was this an educational intervention???
  8. Line 171:  ' quality of advice assessed comprehensively'  . Was this assessment by the Mystery Shoppper?
  9. See 6 above:   was this assessment subjective?
  10. Discussion:  Line 198. Main effect more efficient........Effects can be at patient/ professional( job satistaction)/health service (quality of care/decreased wastage/reduced risk) levels.
  11. English:  Line 228.   It is the pharmacist's duty...
  12. No mention of issues that could arise if patient prescribed two inhalers that require varying administration techniques.
  13. Not clear if pharmacists can prescribe or dispense some inhalers  without a prescription. Every reader will not be familiar with legal situation in Poland.
  14. Lines 280 -----  as pharmacists most accessible health care profession they are pivotal in sign posting physicians/ respiratory specialist/physiotherapists etc. to tackle issue of underdiagnosis (Line 277).
  15. Conclusion not clear enough. See lines 305-308
  16. English lines 309-311 needs to be improved.
  17. Lines  306-308  clarify content actually based on results
  18. Limitations  Line 319........this study...improve English.

Well done

Author Response

Dear Reviewer

I would like to kindly inform you that we have made the necessary corrections, as recommended by the reviewers. The corrections made are listed in the table below.

If you have any further questions don't hesitate to contact me.

Regards,
Weronika Guzenda

Reviewer 2 Report

This was an interesting paper of relevance to those working in community pharmacy / training community pharmacists as well as wider healthcare team. Poor use of inhalers is an important topic and warrants raised awareness about how to improve this. In particular this study looks at the topic in a context whereby there is currently limited support / education around this.

The introduction set the scene well however the aim says a SOP was implemented and assessed - was this really the case or was it more that the trained pharmacists were trained to an SOP and then you assessed how well they educated patients afterwards? The aims in the abstract say the latter. Based on the information provided we don't know that the trained pharmacists actually used the SOP to educate the patients or whether they were just more aware of what should be covered because of the training (which used SOPs). This is an important distinction. If you were assessing use of SOPs this needs to be made explicit. If not then you need to be more cautious in your conclusions as you can't say that SOPs improve things, simply that training on the topic (including use of SOPs) improves things. I think your abstract, which focuses on education rather than SOPs is a more accurate reflection of what you assessed and the conclusions you can draw.

I found the methods quite hard to understand. This would warrant from a review and clarification / re-ordering of some information, in particular:

How were the numbers 150 (total) and 43/107 (two groups) arrived at? Were they based on convenience or were calculations undertaken? Was it intentional to have a much larger control group?

It was quite confusing how the narrative jumped from the MS to the training then back to the MS and it took a couple of reads to work out what was going on. Maybe start by describing the training then introduce the MS study and how you recruited 43 from the 240 trained individuals plus the other 107 non-trained individuals. Add how these 107 were identified and recruited. 

I found the results easy to follow. There was perhaps a bit too much repetition between the tables and narrative but this was not problematic - I found it easy to interpret the tables.

Depending on journal instructions, limitations would make more sense before the conclusions. It might more appropriately be titled ' strengths and limitations' as both are covered. Reference ought to be made to the self-selection of pharmacists to the training - your trained group might therefore be generally more engaged and keen to educate patients well vs your untrained group and it is hard to know how much of the results was down to this difference between the groups. Would giving SOPs to people who are not keen to do training and improve knowledge make a difference?

It would also be good to see some commentary in the discussion around how easy it would be to expand this since the training seemed quite in-depth - is it more widely reproducible? I also didn't understand the point being made in line 306-308. (see notes above re SOPs vs education in regard to the conclusions).

There were a few minor referencing / typographical etc. issues to review:

Line 65-66 ('many problems are caused...') and line 68-69 ('very often...') both need citations adding.

Line 142  typo (od vs of)

Line 153 'was irrelevant' is slightly misleading as we don't know how they viewed it, simply that they didn't cover it so maybe 'not asked' would be more appropriate.

Line 156-7 3 mins is not almost half as long as 5.5 mins - just over half as long might be clearer.

Line 171 use of the word 'comprehensively' is not appropriate given this is described as a subjective impression score in methods. Maybe replace with 'globally' or 'subjectively'

Lines 190-193 - instructions re producing discussion have been left in and need to be removed.

Line 319 'his' should read 'this'

Author Response

Dear Reviewer

I would like to kindly inform you that we have made the necessary corrections, as recommended. The corrections made are listed in the table below.

If you have any further questions don't hesitate to contact me.

Regards,
Weronika Guzenda
